# Changes in Subjective Cognitive and Social Functioning in Parkinson’s Disease from Before to During the COVID-19 Pandemic

**DOI:** 10.3390/healthcare13010070

**Published:** 2025-01-02

**Authors:** Nishaat Mukadam, Shraddha B. Kinger, Sandy Neargarder, Robert D. Salazar, Celina Pluim McDowell, Juliana Wall, Rini I. Kaplan, Alice Cronin-Golomb

**Affiliations:** 1Department of Psychological and Brain Sciences, Boston University, 900 Commonwealth Ave., 2nd Floor, Boston, MA 02215, USA; mukadamn@bu.edu (N.M.); shradk@bu.edu (S.B.K.); sneargarder@bridgew.edu (S.N.); rdsalaz@bu.edu (R.D.S.); cpluim@bu.edu (C.P.M.); kaplanri@bu.edu (R.I.K.); 2Department of Psychology, Bridgewater State University, Bridgewater, MA 02324, USA

**Keywords:** subjective cognition, social functioning, COVID-19, gender, Parkinson’s disease

## Abstract

**Background/Objectives:** Social isolation and health-related consequences of the COVID-19 pandemic may have significantly impacted quality of life in people with Parkinson’s disease (PwPD). The effect of the COVID-19 pandemic specifically on subjective cognition and social functioning in PwPD is poorly understood. We conducted a longitudinal analysis of changes in subjective cognitive and social functioning in PwPD before (T1, 2017–2019) and during (T2, 2021) the COVID-19 pandemic. **Methods:** At T1, 347 PwPD completed online surveys. At T2, 123 of them (54 males, 69 females) responded to follow-up questionnaires including Quality of Life in Neurological Disorders (Neuro-QoL) subscales, Beck Depression Inventory-II, Parkinson’s Anxiety Scale, motor and non-motor experiences of daily living from the MDS-Unified Parkinson’s Disease Rating Scale, and the Coronavirus Impact Scale. **Results:** T1–T2 declines in subjective cognition and social functioning both were correlated with more anxiety, fatigue, and motor symptoms. Additionally, declines in subjective cognition correlated with depression, and with decline in social functioning. Women reported greater COVID-19 impact than men, unrelated to cognition and social functioning; in men, personal experience with COVID-19 was associated with decline in subjective cognition. **Conclusions:** Our finding that subjective cognition and social functioning are associated with different motor and non-motor symptoms of PD suggests that the impacts of PD on subjective cognition and social functioning are complex, which has important implications for treatment.

## 1. Introduction

Parkinson’s disease (PD) is a progressive neurodegenerative disorder characterized by motor symptoms, including resting tremor, gait disturbances, postural instability, bradykinesia, and rigidity [1]. The hallmark motor symptoms are often accompanied by debilitating non-motor symptoms including sleep disorders, gastrointestinal disturbances, mood disturbances, and cognitive dysfunction [2,3]. Both motor and non-motor symptoms of PD interfere with daily functioning, influence the quality of life of persons with PD (PwPD) [4], and can lead to feelings of loneliness and isolation in these individuals [5]. There are a number of factors that contribute to social isolation, social withdrawal, and loneliness in PD, including perceived stigma associated with PD [6,7,8,9], difficulty speaking and communicating [5], feelings of embarrassment as a result of symptoms such as tremor, dyskinesia, drooling, difficulty eating and drinking, poor balance, freezing of gait, and fear of falling [10,11]. Feelings of social withdrawal, social isolation, and loneliness further impact the quality of life of PwPD [5,12,13].

In the general population, restrictions and lockdowns imposed during the COVID-19 pandemic contributed to social isolation and loneliness [14], which are known to negatively impact physical, mental and emotional health [15]. Social isolation is particularly harmful to older adults with concomitant poor health, sensory impairments that interfere with participation in social activities, and a social network that becomes smaller through death of family, friends, and acquaintances in their age cohort [16,17]. In PwPD, the motor and non-motor symptoms contribute to social isolation and loneliness above and beyond the contributions associated with aging and COVID-19, suggesting that the effects of the pandemic may have been particularly salient for PwPD [5]. Further, social isolation has a negative effect on cognitive functioning [18,19] and is likely to result in cognitive decline [20,21,22]. PD often diminishes social and cognitive functioning [11]; the pandemic may have exacerbated these symptoms.

The purpose of the present study was to evaluate the effects of living through the COVID-19 pandemic on subjective cognitive and social functioning in PwPD. We conducted a two-timepoint (T1 and T2) longitudinal analysis of changes in quality of life, particularly cognitive and social functioning, from before to during the pandemic. We expected declines in subjective cognition and social functioning, and that these declines would relate to personal difficulties associated with COVID-19 and to the passage of time. Measures included mood, fatigue, well-being, and aspects of motor and non-motor function, as these may impact cognition and social functioning. Further, we conducted analyses separately by gender because of research indicating that men and women with PD may have been differentially impacted by the pandemic, with women exhibiting more symptoms of depression, anxiety, and stress, lower frustration tolerance, higher levels of distress, more difficulty relaxing, and poorer sleep quality [23]. In general, PD has been found to affect men and women differently regarding mood and cognition [24,25].

## 2. Materials and Methods

We conducted a two-timepoint longitudinal study evaluating changes in subjective cognitive and social functioning in PwPD from before to during the COVID-19 pandemic. The first time-point (T1) was associated with participation by PwPD in the Boston University Online Survey Study of Parkinson’s Disease (BOSS-PD) between June 2017 and December 2018 (see Islam et al. [8]). Inclusion criteria for the original study were diagnosis of PD without dementia, 40+ years of age, 8+ years of education, proficient in English, and access to a computer. Exclusion criteria were active neoplasm, serious cardiac disease, other serious chronic medical illness, prior intracranial surgery, history of traumatic brain injury, psychiatric or neurological diagnoses other than PD, history of alcoholism or other drug abuse or treatment with electroconvulsive therapy. During the pandemic, between March and June 2021, the sample was recontacted for the follow-up PD-COVID survey (T2) (see Kinger et al. [26]) to assess how the cohort was coping during the pandemic. We received responses from 130 of the original BOSS-PD participants. Data for seven individuals who did not complete all required measures were excluded, resulting in a final sample of 123 participants. Consent was obtained before survey access. All procedures were approved by the Institutional Review Board of Boston University and were in accordance with the Helsinki Declaration.

### 2.1. Participants

Participants in BOSS-PD were recruited through Fox Trial Finder, Clinicaltrials.gov, American Parkinson Disease Association websites, and other community outreach. BOSS-PD included 347 PwPD (165 men, 180 women, 2 not indicated). Mean age was 64.8 years (SD = 8.5), years of education was 16.8 years (SD = 2.6), and disease duration was 5.4 years (SD = 4.6). We attempted to recontact all BOSS-PD participants for the follow-up PD-COVID study, and received responses from 130 of the original 347 participants. Data from seven participants who left study-relevant questionnaires incomplete at T2 were excluded. The final sample with both T1 and T2 data comprised 123 PwPD. We compared this PD-COVID sample of 123 with the remaining 217 participants of BOSS-PD who did not respond at T2 to establish how representative it was of the original sample.

### 2.2. Measures

BOSS-PD and PD-COVID were both online Qualtrics surveys consisting of demographic/clinical information and self-report questionnaires on motor and non-motor symptoms of PD and quality of life. Measures relevant to the current study are described below. All measures except for the Coronavirus Impact Scale were included at both timepoints.

Coronavirus Impact Scale (CIS; T2 only) [27]: This scale was used to measure pandemic-related impact. The CIS is a 12-item scale that assesses the extent of change in routines (item #1), family income/employment (item #2), access to food (item #3), medical healthcare access (item #4), mental healthcare access (item #5), social supports (item #6), pandemic-related stress (item #7), family stress and discord (item #8), personal COVID-19 diagnosis (item #9). The first 9 items of the scale are rated on a 4-point scale with higher scores indicating greater impact: 1 = none/no change, 2 = mild, 3 = moderate, 4 = severe. Item #10 asks about number of people diagnosed in the immediate family, and item #11 asks about number of people diagnosed in the extended family and close friends. Item #12 is open-ended, asking for other ways in which the COVID-19 pandemic has impacted these individuals. A CIS total score is calculated by totaling responses of items #1 through #9.

Quality of Life in Neurological Conditions (Neuro-QoL) [28]: These are measures of health-related quality of life in multiple domains. Items in each domain are rated on a scale from 1 (none/never) to 5 (cannot do/always). A total score is obtained for each of the domains of the Neuro-QoL by totaling item responses in each domain. We administered the following scales of the Neuro-QoL:*Social Roles and Activities (Neuro-QoL_social_):* This is an 8-item scale that measures an individual’s degree of involvement in social roles, activities, and responsibilities, including work, family, friends and leisure. Total scores range from 8 to 40, with higher scores indicating better performance.*Cognitive Function (Neuro-QoL_cog_):* This is an 8-item scale that measures an individual’s self-reported difficulties in various cognitive abilities and their application, including memory, attention, decision making, planning, organization, calculating, remembering, and learning. Total scores range from 8 to 40, with higher scores indicating better performance.*Positive Affect and Well-Being (Neuro-QoL_well-being_):* This is a 9-item scale that measures individual’s sense of well-being, life-satisfaction, and an overall sense of purpose and meaning. Total scores range from 9 to 45, with higher scores indicating better performance.*Fatigue (Neuro-QoL_fatigue_):* This is an 8-item scale that measures degree of fatigue ranging from tiredness to an overwhelming and sustained sense of exhaustion. Total scores of this scale range from 8 to 40, with higher scores indicating greater fatigue.

The measures of primary interest for this study were Social Roles and Activities, and Cognitive Functioning.

Movement Disorders Society-Unified Parkinson’s Disease Rating Scale (MDS-UPDRS) [29]: This is a four-part measure of severity of motor and non-motor symptoms of PD. We included parts Ib and II, being unable to conduct part Ia, the clinician-rated portion, and part III, the motor exam, in this online study. Items in each section are rated on a scale from 0 to 4. Total scores are obtained for each section by adding responses in that section.

*Part I—Non-Motor Aspects of Experiences of Daily Living (UPDRS_non-motor_):* This is a 7-item self-report questionnaire that measures various non-motor aspects of PD including sleep problems, daytime sleepiness, pain, urinary problems, constipation, lightheadedness, and fatigue. Total UPDRS_non-motor_ scores can range from 0 to 28, with higher scores indicating more severe symptoms.*Part II—Motor Aspects of Experiences of Daily Living (UPDRS_motor_):* This is a 13-item self-report questionnaire on motor-symptoms, including problems with speech, drooling, swallowing, eating, dressing, hygiene management, handwriting, engaging in hobbies and other activities, turning in bed, tremor, getting out of a bed, car or chair, walking and balance, and freezing of gait. Total UPDRS_motor_ scores can range from 0 to 52, with higher scores indicating more severe symptoms.

Beck Depression Inventory-II (BDI-II) [30]: This is 21-item self-report questionnaire that assesses severity of depressive symptoms on a scale of 0 to 3. Total scores range from 0 to 63, with higher scores indicating more severe depression symptoms.

Parkinson’s Anxiety Scale (PAS) [31]: This is a 12-item self-report scale measuring the severity of anxiety in PwPD. Each item is scored on a 5-point scale, ranging from 0 (not at all/never) to 4 (severe/almost always). The maximum score for this scale is 48. Higher scores indicate more severe anxiety symptoms.

### 2.3. Statistical Analyses

Two-sample Wilcoxon signed rank tests (two-tailed) were used to examine changes from T1 to T2 on the Neuro-QoL scales, BDI-II, PAS and the UPDRS motor and non-motor scores (alpha 0.05). Difference scores (T2 subtracted from T1) were calculated for all tests except CIS (administered at T2 only), as well as for demographic and clinical variables. Pearson correlations were conducted between demographic and clinical variables, CIS total score, and T1–T2 difference scores for all measures and the two variables of interest, i.e., Neuro-Qol_social_ and Neuro-QoL_cog_ (alpha 0.01 to account for multiple comparisons). Variables that were significantly correlated with changes in social roles and activities and cognitive functioning were entered into two linear regression models—one with changes in social roles and activities as the outcome variable, and the other with changes in cognitive function as the outcome variable. To evaluate gender differences, we first conducted independent samples *t*-tests at T1 and T2 to evaluate whether there were differences, before or during the pandemic, between men and women in clinical/demographic variables or self-reported motor and non-motor symptoms. Exploratory analyses of T2 data using independent samples *t*-tests investigated whether men and women were differentially impacted by the pandemic, per the CIS total score and item scores. CIS item analysis was conducted using Spearman correlations to examine the relative impact of pandemic-related factors on subjective cognition and social functioning in men and women separately (alpha 0.05). Data were analyzed using SPSS 27.0.1.0 statistical software (SPSS, IBM Corp, Chicago, IL, USA).

## 3. Results

Participant demographic characteristics appear in Table 1. A total of 123 PwPD participated at both T1 and T2 (54 men, 69 women, age range 44–84 at T2, number of years of education range 10–21 years). A comparison of the “Only BOSS-PD” sample (that did not participate in PD-COVID) to those who also participated in PD-COVID revealed that at T1, the Only BOSS-PD sample reported significantly worse symptoms on UPDRS_motor_ (t(314.62) = 2.7, *p* = 0.008), UPDRS_non-motor_ (t(338) = 2.4, *p* = 0.02), Neuro-QoL_cog_ (t(338) = 2.2, *p* = 0.03), Neuro-QoL_fatigue_ (t(338) = 2.6, *p* = 0.01), and PAS (t(299.16) = 2.3, *p* = 0.03). There was also a trend for the Only BOSS-PD sample to be older (t(335) = 1.9, *p* = 0.05) and to report worse symptoms on the BDI-II (t(338) = 2.0, *p* = 0.05).

### 3.1. Whole Sample

Results for the remaining analyses apply to the sample of 123 PwPD seen at both T1 and T2. We first discuss the whole sample, and then analyses by gender separately.

COVID-19 impact: For the whole sample, CIS total scores ranged from 9 to 30. Scores on the question about personal diagnosis of COVID-19 ranged from 1 to 3, with 113 endorsing no effects, 8 mild, 2 moderate, and 0 severe effects.

T1–T2 change: For the whole sample, significant declines from T1 to T2 were seen only on Neuro-QoL_social_ (z = 3.6, *p* < 0.001), Neuro-QoL_cog_ (z = 3.6, *p* < 0.001), and UPDRS_motor_ (z = 9.6, *p* < 0.001). There were no changes for any other variables (Table 2).

Correlation and Regression Analyses: Difference scores between T1 and T2 were calculated for all measures. These difference scores, as well as demographic and clinical variables, including age, education, disease duration at T2, and CIS scores at T2 were correlated with Neuro-QoL_social_ and Neuro-QoL_cog_ (Table 3). Regression analyses were conducted to assess the effects of variables that correlated significantly with change in social roles and activities and change in cognitive functioning. All assumptions for the linear regression model were met. Age, education, and CIS score at T2 did not correlate significantly with measures of subjective cognition (Neuro-QoL_cog_) or social functioning (Neuro-QoL_social_) (all r’s ≤ 0.10, all *p*’s > 0.05).

*Subjective cognition.* T1–T2 change on Neuro-QoL_cog_ correlated negatively with UPDRS_motor_ (r = −0.23, *p* = 0.01), BDI-II (r = −0.44, *p* < 0.001), PAS (r = −0.40, *p* < 0.001), and Neuro-QoL_fatigue_ (r = −0.33, *p* < 0.001), with a trend for disease duration (r = −0.19, *p* = 0.04). In the regression model (Table 4), change in Neuro-QoL_cog_ was regressed on disease duration at T2 and changes in UPDRS_motor_, BDI-II, PAS, and Neuro-QoL_social_, Neuro-QoL_fatigue_ scores. Change in cognitive function was significantly predicted by disease duration at T2 (B = −0.14, t = 2.0, *p* < 0.05), and by changes in BDI-II (B = −0.21, t = 3.1, *p* = 0.002), PAS (B = −0.19, t = 2.3, *p* = 0.02), and Neuro-QoL_social_ (B = 0.12, t = 2.2, *p* = 0.03). The overall model was significant and the predictors explained 29% of the variance after adjusting for the number of predictors in the model (adjusted r^2^ = 0.29; F = 9.3, *p* < 0.001).

*Social functioning.* T1–T2 change on Neuro-QoL_social_ correlated negatively with UPDRS_motor_ (r = −0.32, *p* < 0.001), PAS (r = −0.28, *p* = 0.002), and Neuro-QoL_fatigue_ (r = −0.26, *p* = 0.003), with trends for BDI-II (r = −0.18, *p* = 0.05) and NeuroQoL_well-being_ (r = 0.22, *p* = 0.02). In the regression model (Table 5), change in Neuro-QoL_social_ was regressed on changes in UPDRS_motor_, PAS, Neuro-QoL_cog_, and Neuro-QoL_fatigue_ scores. Of these variables, changes in UPDRS_motor_ (B = −0.31, t = 2.38, *p* = 0.02) and Neuro-QoL_cog_ (B = 0.35, t = 2.3, *p* = 0.02) were significant predictors of social roles and activities. After adjusting for the number of variables in the model, this model accounted for 17% of the variance in social roles and activities (adjusted r^2^ = 0.17, F = 5.9, *p* < 0.001).

### 3.2. Gender-Wise Analyis

Gender effects: Comparing gender differences between self-reported motor and non-motor symptoms at both T1 and T2, we found that there were no significant differences between men and women with regard to their reported motor symptoms, cognition, social functioning, well-being, fatigue, depression and anxiety (all *p*’s > 0.05). At T2, however, women had longer disease duration than men (t[121] = 2.2, *p* = 0.03; M_men_ = 7.1 [SD = 3.4], M_women_ = 9.0 [SD = 5.6]) and reported greater overall impact of COVID-19 (CIS_total_: t[121] = 2.2, *p* = 0.03; M_men_ = 15.7 [SD = 3.6], M_women_ = 17.0 [SD = 3.1]), specifically related to food access (CIS_item3_: t[112.4] = 2.3, *p* = 0.02; M_men_ = 1.1 [SD = 0.29], M_women_ = 1.3 [SD = 0.50]) and pandemic-related stress (CIS_item7_: t[121] = 3.0, *p* = 0.003; M_men_ = 1.7 [SD = 0.69], M_women_ = 2.1 [SD = 0.69]). Figure 1 provides item-wise responses to the first nine CIS items by gender.

Correlation Analyses for Gender: Gender-wise correlation analyses can be found in Table 6 and Table 7. For women, change in Neuro-QoL_cog_ from T1 to T2 correlated with changes in NeuroQoL_social_, Neuro-QoL_fatigue_, and BDI-II (all r’s ≥ 0.31, all *p*’s < 0.01), with trends for UPDRS_motor_ and PAS (both r’s ≥ 0.25, both *p*’s < 0.05). Change in NeuroQoL_social_ correlated with changes in Neuro-QoL_fatigue_, Neuro-QoL_well-being_, UPDRS_motor_, and PAS (all r’s ≥ 0.33, all *p*’s ≤ 0.01), with a trend for UPDRS_non-motor_ (r = −0.27, *p* = 0.03). For men, changes in Neuro-QoL_cog_ correlated with changes in Neuro-QoL_fatigue_, BDI-II, and PAS (all r’s ≥ 0.31, all *p*’s ≤ 0.01). Age and education did not correlate with changes in Neuro-QoL_social_ or Neuro-QoL_cog_ in either men or women (all r’s ≤ 0.16, all *p*’s > 0.01).

To examine whether there were significant differences between men and women in the size of the correlations, we conducted Fisher-z transformations. There were significant differences for Neuro-QoL_social_ and Neuro-QoL_well-being_ (z = 1.72, *p* = 0.04) and NeuroQoL_social_ and UPDRS_motor_ (z = 1.99, *p* = 0.02). There was also a trend towards a gender difference in the correlation between Neuro-QoL_social_ and UPDRS_non-motor_ (z = 1.59, *p* = 0.06), and between Neuro-QoL_cog_ and PAS (z = 1.56, *p* = 0.06).

CIS Item Analysis by Gender: We conducted Spearman correlations separately for men and women between (1) CIS_total_ and (2) individual CIS items, with PD variables and scores on Neuro-QoL_cog_ and Neuro-QoL_social_ (Table 8 and Table 9).

*Men.* (1) CIS_total_. In men, CIS_total_ significantly correlated with age at T2, with higher impact associated with younger age (*ρ* = −0.23, *p* = 0.02). (2) CIS items. Score decline on Neuro-QoL_cog_ from T1 to T2 correlated with two CIS items—access to mental health treatment, and personal COVID-19 diagnosis (both *ρ*’s ≥ 0.30, both *p*’s ≤ 0.03).

*Women.* In women, there were no significant correlations between CIS_total_ or individual CIS items with PD variables or Neuro-QoL_cog_.

For Neuro-QoL_social_, no correlations were found between CIS_total_ or individual CIS items for either men or women (all *ρ* ≤ 0.17, all *p* > 0.05) (Table 6 and Table 7).

## 4. Discussion

This study assessed the impact of the COVID-19 pandemic on subjective cognitive and social functioning in PwPD. In line with our predictions, we found that self-reported social and cognitive functioning declined from T1 to T2, although these declines did not correlate with ratings on the Coronavirus Impact Scale. Additionally, we found gender differences in the impact of the COVID-19 pandemic, with women reporting more significant impact from the pandemic than men. There were also gender differences in the impact of COVID-19 on cognitive functioning.

The hypothesis that there would be a decline in both self-reported social functioning and cognitive functioning was supported. The decline in cognitive functioning over time was consistent with findings by Marras et al. [32], who saw declines in self-reported cognition on the NeuroQoL_cog_ in PwPD over a three-year period that was likely not confounded by the pandemic, as the dates of data collection were not reported and the paper was published in 2021. Additionally, there was also worsening of motor symptoms from T1 to T2. These changes in self-reported cognition and social functioning and in motor symptoms could reflect normal disease progression, direct impacts of the COVID-19 pandemic, or indirect effects of the pandemic period such as physical and social restrictions on daily life. We explore each of these as possibilities for the observed results.

Changes in subjective cognitive and social functioning in the whole sample were not found to be related to personal experience with COVID-19, as indicated by CIS scores. Although this finding might lead us to conclude that COVID-19 did not impact cognitive and social functioning in PwPD, the impact of COVID-19 on these aspects of functioning cannot be ruled out entirely. First, our sample reported overall mild personal experience with COVID-19, as indicated by the CIS scores. Like in many research studies, our sample was well-educated and possibly of high enough socioeconomic status to be protected against the more dire consequences of COVID-19 [33,34]. During the pandemic, media coverage resulted in widespread awareness about the devastating impacts of the pandemic on individuals and communities across the globe. Therefore, even though COVID-19 might have impacted our sample significantly in several ways, the milder CIS scores might reflect comparisons our sample may have made between the pandemic’s impact on their own lives compared to people who experienced more severe consequences of the pandemic. In that sense, the CIS responses may not necessarily reflect the true impact of COVID-19 on these individuals. Second, other than the changes seen in cognition and social functioning and in motor symptoms, there was no significant change from T1 to T2 in our sample for any other PD/clinical variables measured. This is not in line with the expected non-motor symptom declines seen in the progression of PD over the course of 3–4 years [35,36]. Therefore, general progression of the disease and/or other factors such as fatigue do not fully explain the declines that were seen in social functioning and cognition. Third, although literature in this field is limited, other studies have shown pandemic-related declines in cognitive functioning and its relation to social functioning in older adults [37,38] and PwPD [39]. Amieva et al. [37] showed a more rapid decline in cognitive functioning in older adults in the few months following the pandemic compared to the slow and non-significant decline over several years prior to the pandemic. These findings provide additional support for the impact of the COVID-19 pandemic on cognitive and social functioning. Taking together the findings from these studies about the negative impact of the pandemic on cognitive functioning of PwPD and older adults, along with the incomplete explanation that PD-related general disease progression provides for the decline in cognitive and social functioning in our sample, it seems likely that the declines in cognitive symptoms seen in our study are likely related to and/or exacerbated by the pandemic.

*Cognitive Functioning:* Our finding that increased depression and anxiety from T1 to T2 predicted decline in subjective cognition is consistent with prior work reporting the relation between depression and anxiety and subjective cognitive concerns [40,41]. This finding also aligns with reported associations between mood and objectively-measured cognition during COVID-19 in PwPD [42]. Further, consistent with existing PD literature on objectively measured cognition [43], we also found that worsened motor symptoms from T1 to T2 correlated significantly with an increase in cognitive concerns. Lastly, as expected, we also found that longer disease duration predicted worse cognitive function.

*Social Functioning:* We found that increases in motor problems predicted worsened social functioning, aligning with prior findings that movement-related issues, facial masking, speech difficulties, and low speech volume interfere with social functioning in PwPD [44]. It is also possible, however, that lower socialization due to the pandemic resulted in less engagement in physical activity, which contributed to the decline in motor function. Further, our finding that increased anxiety, fatigue, and cognitive concerns correlated with poorer social functioning in PwPD is in line with prior work that has focused on factors that lead to social withdrawal in PD [45]. Consistent with prior literature on the impacts of social isolation on cognitive functioning [15,46], our study showed a significant correlation between subjective cognitive and social functioning.

*Gender differences in the impact of COVID-19:* As discussed above, in the whole sample, changes in subjective cognitive and social functioning were not related to personal experience with COVID-19 as indexed by the CIS. Of interest, however, is that we found gender differences in the impact of COVID-19. Women reported greater impact of COVID-19 than men, including experiencing more stress; in men, greater impact (i.e., personal diagnosis) of COVID-19 was related to worse subjective cognition. Worse subjective cognition also correlated significantly with fatigue, depression, poorer social functioning (and trend for motor symptoms and anxiety) in women, and with fatigue, anxiety, depression (and trend for motor symptoms) in men. Poorer social functioning correlated with more severe motor impairment, fatigue, poorer well-being, anxiety, worse subjective cognition (and trend for non-motor severity) in women, with no significant correlations for men. These results suggest that COVID-19 had different impacts on men and women with PD including in subjective cognitive and social functioning.

Overall, our findings suggest that the declines in social and cognitive functioning in PD from before to during the COVID-19 pandemic can be attributed, at least in part, to the actual impact of the pandemic. This finding adds to the existing albeit limited literature on the impact of the pandemic on cognition, depression, anxiety, and social isolation in PwPD [39,47,48]. Several factors contribute to worsening of social and cognitive functioning in PD, including motor symptoms, depression, anxiety, fatigue, and disease duration. Social isolation and poor cognitive functioning have a significant impact on the quality of life of PwPD [5,13,49,50]. Management of these factors is important to improve quality of life in these individuals. This can be addressed through direct treatment of cognitive difficulties (e.g., pharmacological intervention, cognitive rehabilitation) and social isolation (increased opportunities for social interaction), as well as addressing factors that are contributing to poor social and cognitive functioning (e.g., depression, anxiety, fatigue). Our findings also suggest that men and women with PD were impacted in different ways by the pandemic, with some differences in factors that impacted social and cognitive functioning. This can help tailor more effective treatments for men and women.

This study had several strengths: (i) Online assessment enabled us to collect data from the same participants before and during the COVID-19 period when in-person studies were precluded, and to collect data quickly, so that for PD-COVID, we assessed all PwPD at the same short time period of the pandemic. (ii) The heterogeneity of PD was reflected in some aspects of demographics and clinical characteristics, including age, age at onset, duration, and severity. (iii) Women outnumbered men in both samples, whereas many in-lab studies of PD enroll significantly more men. (iv) Prior work [51] showed that COVID-19 infection worsened both motor (e.g., stiffness, tremor, gait) and non-motor (e.g., mood, cognition, fatigue) symptoms in PD. Given that our sample largely consisted of individuals who had not had a personal diagnosis of COVID-19, as indicated by responses to CIS-item 9 (personal diagnosis of COVID-19), we were able to assess the impact of the pandemic mainly without the influence of the COVID-19 infection.

### Limitations

This study was subject to limitations. First, the sample was not representative of the general population, consisting of primarily white, non-Hispanic, highly educated PwPD, with computer and internet access, who experienced mostly mild effects of the pandemic as evidenced by scores on the CIS. Second, at T1, the sample that had participated only in BOSS-PD, compared to the BOSS-PD sample that also participated in PD-COVID, reported significantly worse motor and non-motor symptoms, subjective cognition, fatigue, and anxiety, and a trend toward being older and having more depressive symptoms, suggesting a survivor effect that limited the range of severity in our PD-COVID sample [6,8]. A survivor benefit was also suggested by disease duration being associated with improved subjective cognition; alternatively, individuals with lower cognitive functioning may have lacked metacognitive awareness about their true cognitive abilities. BOSS-PD participants who were more affected by the pandemic may have been unable to respond to the PD-COVID survey due to significant health issues. Future research should focus on PwPD who were affected more strongly by the pandemic. Another limitation is one inherent to many online-survey studies: all aspects of functioning were by self-report only, including cognition. It would have been valuable to assess cognition objectively to provide context for evaluating subjective cognition and other aspects of function. Our study did not measure loneliness, which is another limitation, given the potential impacts of loneliness in PwPD during the pandemic [48]. Finally, the PD-COVID survey was conducted in 2021, after vaccinations had begun and when restrictions were starting to be lifted, which likely resulted in lower pandemic impact on quality of life than in the months prior to the survey.

## 5. Conclusions

Overall, our findings indicate that the impact of PD on subjective cognition and social functioning is complex and was further complicated by the pandemic. Therefore, in addition to motor symptoms, treatment of PwPD should take into account these factors, as well as the role that gender plays in subjective cognition and social functioning. Future studies should focus on broader samples in regard to demographics and pandemic impact, include in-person as well as online survey samples, and evaluate change using objective measures of function.

## Figures and Tables

**Figure 1 healthcare-13-00070-f001:**
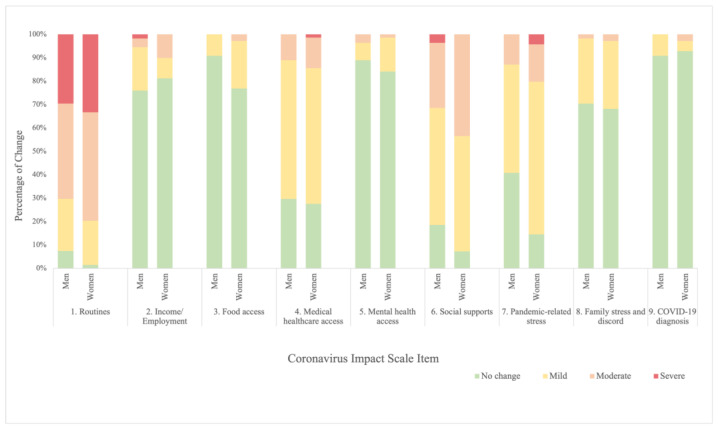
Item-wise ratings of item #1–item #9 on the Coronavirus Impact Scale (CIS) by Gender.

**Table 1 healthcare-13-00070-t001:** Participant characteristics for all BOSS-PD participants (2017–2018), BOSS-PD only participants (2017–2018), matched BOSS-PD and PD-COVID participants at T1 (2017–2018) and matched BOSS-PD and PD-COVID participants at T2 (2021).

	All BOSS-PD (2017–2018) *n* = 347	Only BOSS-PD (2017–2018) *n* = 217	Matched BOSS-PD + PD-COVID at T1 (2017–2018) *n* = 123 ^a^	Matched BOSS-PD + PD-COVID at T2 (2021) *n* = 123 ^a^
	Mean (SD)	Mean (SD)	Mean (SD)	Mean (SD)
Age	64.8 (8.5)	65.6 (8.7)	63.8 (7.9)	67.2 (7.9)
(years)		*n* = 214		
Min.	40	45	40	44
Max.	91	91	81	84
Education	16.7 (2.6)	16.7 (2.7)	16.8 (2.4)	16.8 (2.4)
(years)		*n* = 214		
Min.	10	10	10	10
Max	21	21	21	21
Gender (N)				
Male	165	108	54	54
Female	180	107	69	69
No response	2	2	0	
PD Duration (years)	5.4 (4.6)	5.6 (4.5)	5.0 (4.8)	8.1 (4.8)

All BOSS-PD = Entire original sample of 347 at T1; Only BOSS-PD = Participants at T1 who did not participate at T2; Matched BOSS-PD + PD-COVID = Participants at both T1 and T2. ^a^ Number of participants in PD-COVID (130) minus 7 who did not complete all measures.

**Table 2 healthcare-13-00070-t002:** Results on study measures with comparison between matched BOSS-PD + PD-COVID participants at T1(2017–2018) and T2 (2021).

	All BOSS-PD *n* = 347	Only BOSS-PD *n* = 217	Matched BOSS-PD + PD-COVID at T1 *n* = 123	Matched BOSS-PD + PD-COVID at T2 *n* = 123	T1–T2 Mean Change (SE)	Two-Sample Wilcoxon Signed Rank Test Between Matched Participants at T1 and T2 (z-Score)
Neuro-QoL_social_ ^a^ (score range: 8–24)	34.5 (6.9)	34.2 (7.1)	35.4 (6.1)	33.8 (6.6)	1.6 (0.6)	3.6 **
Neuro-QoL_cog_ ^a^ (score range: 8–24)	33.5 (6.4)	32.9 (6.8)	34.5 (5.6)	33.0 (6.1)	1.5 (0.4)	3.6 **
Neuro-QoL_fatigue_ (score range: 8–24)	17.4 (7.0)	18.2 (7.1)	16.1 (6.8)	16.8 (6.6)	−0.7 (0.5)	2.0
Neuro-QoL_well-being_ (score range: 9–26)	36.3 (7.8)	35.9 (8.1)	37.1 (7.3)	36.1 (7.4)	1 (0.6)	1.8
UPDRS-I_non-motor_ (score range: 0–28)	8.0 (4.3)	8.3 (4.5)	7.2 (3.8)	7.6 (3.8)	−0.4 (0.3)	1.6
UPDRS-II_motor_ (score range: 0–52)	10.6 (7.5)	11.3 (8.0)	9.2 (6.0)	11.5 (6.8)	−2.3 (0.4)	9.6 **
BDI-II ^b^ (score range: 0–63)	8.6 (6.8)	9.0 (6.8)	7.5 (6.5)	8.0 (6.8)		0.52
PAS ^b^ (score range: 0–48)	6.6 (6.8)	7.2 (7.2)	5.6 (5.8)	5.5 (6.3)	0.1 (0.4)	0.77
CIS Total (score range: 9–36)	-	-	-	16.4 (3.4)	-	-

All BOSS-PD = Entire original sample of 347 at T1; Only BOSS-PD = Participants at T1 who did not participate at T2; Matched BOSS-PD + PD-COVID = Participants at both T1 and T2. ^a^ For Neuro-QoL_social_ and Neuro-QoL_cog_, positive change reflects worse function and negative change reflects better function. ^b^ Clinical cut-off for BDI-II and PAS = 13. ** *p* < 0.001 (significant) Neuro-QoL = Quality of Life in Neurological Conditions: Neuro-QoL_social_ = Social Roles and Activities; Neuro-QoL_cog_ = Cognitive Function; Neuro-QoL_well-being_ = Well-Being; Neuro-QoL_fatigue_ = Fatigue; UPDRS = MDS-Unified Parkinson’s Disease Rating Scale: UPDRS-I_non-motor_ = non-motor; UPDRS-II_motor_ = motor; BDI-II = Beck Depression Inventory-II; PAS = Parkinson’s Anxiety Scale; CIS = Coronavirus Impact Scale.

**Table 3 healthcare-13-00070-t003:** Pearsons correlations among demographic/clinical characteristics at T2, and change scores from T1 to T2 on Neuro-QoL scales, UPDRS scales, BDI-II, and PAS.

	T2 Age	Education	T2 PD Duration	Neuro-QoL_social_	Neuro-QoL_cog_	Neuro-QoL_fatigue_	Neuro-QoL_well-being_	UPDRS_non-motor_	UPDRS_motor_	BDI-II	PAS	CIS
T2 Age	-											
Education	0.11	-										
T2 PD Duration	0.10	−0.19	-									
Neuro-QoL_social_	0.02	0.01	−0.07	-								
Neuro-QoL_cog_	−0.01	0.10	−0.19 *	0.32 ***	-							
Neuro-QoL_fatigue_	−0.02	−0.05	0.07	−0.26 *	−0.33 **	-						
Neuro-QoL_well-being_	0.18 *	−0.05	−0.06	0.22 *	0.10	−0.32 ***	-					
UPDRS-I_non-motor_	−0.07	−0.05	0.05	−0.13	−0.12	0.21 *	−0.04	-				
UPDRS-II_motor_	−0.23 **	−0.14	0.15	−0.32 ***	−0.23 **	0.32 ***	−0.14	0.36 ***	-			
BDI	0.04	−0.15	0.09	−0.18 *	−0.44 **	0.33 ***	−0.30 **	0.09	0.28 **	-		
PAS	−0.20 *	−0.10	−0.03	−0.28 **	−0.40 **	0.30 ***	−0.29 **	0.25 **	0.33 ***	0.43 ***	-	
T2 CIS	−0.27 **	−0.10	0.18	0.02	0.06	0.03	0.11	−0.01	0.15	0.09	−0.03	-

*** *p* < 0.001, ** *p* < 0.01 (significant), * *p* < 0.05 (trend) Note. T1–T2 used for all variables unless specified. Neuro-QoL = Quality of Life in Neurological Conditions: Neuro-QoL_social_ = Social Roles and Activities; Neuro-QoL_cog_ = Cognitive Function; Neuro-QoL_well-being_ = Well-Being; Neuro-QoL_fatigue_ = Fatigue; UPDRS = MDS-Unified Parkinson’s Disease Rating Scale; UPDRS-I_non-motor_ = non-motor; UPDRS-II_motor_ = motor; BDI-II = Beck Depression Inventory-II; PAS = Parkinson’s Anxiety Scale; CIS = Coronavirus Impact Scale.

**Table 4 healthcare-13-00070-t004:** Regression Model 1: Predicting change in *Neuro-QoL_cog_* (T1–T2) from change in UPDRS-II_motor_, BDI-II, PAS, *Neuro-QoL_social_*, *Neuro-QoL_fatigue_* and disease duration at T2.

Variable	B	SE	β	t
Constant	2.38	0.70		3.4 ***
UPDRS-II_motor_	0.04	0.08	0.04	0.5
BDI-II	−0.212	0.07	−0.27	3.1 **
PAS	−0.19	0.08	−0.21	2.3 **
Neuro-QoL_social_	0.12	0.05	0.19	2.2 *
Neuro-QoL_fatigue_	−0.11	0.07	−0.13	1.5
T2 PD Duration	−0.14	0.07	−0.16	2.0 *

*** *p* < 0.001, ** *p* < 0.01 (significant), * *p* < 0.05 (trend) Note. T1–T2 used for all variables unless specified. Neuro-QoL = Quality of Life in Neurological Conditions: Neuro-QoL_social_ = Social Roles and Activities; Neuro-QoL_cog_ = Cognitive Function; Neuro-QoL_well-being_ = Well-Being; Neuro-QoL_fatigue_ = Fatigue; UPDRS = MDS-Unified Parkinson’s Disease Rating Scale; UPDRS-II_motor_ = motor; BDI-II = Beck Depression Inventory-II; PAS = Parkinson’s Anxiety Scale; CIS = Coronavirus Impact Scale.

**Table 5 healthcare-13-00070-t005:** Regression Model 2: Predicting Change in *Neuro-QoL_social_* (T1–T2) from Change in UPDRS-II_motor_, PAS, *Neuro-QoL_cog_*, *Neuro-QoL_fatigue_* and *Neuro-QoL_well-being_*.

Variable	B	SE	β	t
Constant	0.18	0.68		0.3
UPDRS-II_motor_	−0.31	0.13	−0.22	2.4 *
PAS	−0.207	0.144	−0.141	1.4
Neuro-QoL_cog_	0.35	0.15	0.21	2.3 *
Neuro-QoL_fatigue_	−0.09	0.13	−0.07	0.7
Neuro-QoL_well-being_	0.13	0.09	0.13	1.4

* *p* < 0.05 Note. T1–T2 used for all variables unless specified. Neuro-QoL = Quality of Life in Neurological Conditions: Neuro-QoL_social_ = Social Roles and Activities; Neuro-QoL_cog_ = Cognitive Function; Neuro-QoL_well-being_ = Well-Being; Neuro-QoL_fatigue_ = Fatigue; UPDRS = MDS-Unified Parkinson’s Disease Rating Scale; UPDRS-II_motor_ = motor; BDI-II = Beck Depression Inventory-II; PAS = Parkinson’s Anxiety Scale; CIS = Coronavirus Impact Scale.

**Table 6 healthcare-13-00070-t006:** Pearson correlation among demographic/clinical characteristics, scores on Neuro-QoL scales, UPDRS scales, BDI, PAS, and CIS for men.

	T2 Age	Education	T2 PD Duration	Neuro-QoL_social_	Neuro-QoL_cog_	Neuro-QoL_fatigue_	Neuro-QoL_well-being_	UPDRS_non-motor_	UPDRS_motor_	BDI-II	PAS
T2 Age	--										
Education	0.01	--									
T2 PD Duration	0.15	−0.12	--								
Neuro-QoL_social_	−0.05	−0.16	0.14	--							
Neuro-QoL_cog_	−0.02	0.04	−0.18	0.22	--						
Neuro-QoL_fatigue_	−0.05	−0.01	−0.03	−0.12	−0.36 **	--					
Neuro-QoL_well-being_	0.27 *	−0.09	−0.02	0.08	0.04	−0.24	--				
UPDRS-I_non-motor_	−0.06	0.05	0.19	0.02	−0.16	0.28 *	0.04	--			
UPDRS-II_motor_	−0.20	−0.03	0.01	−0.15	−0.22	0.35 **	0.06	0.40 **	--		
BDI-II	0.13	−0.07	0.04	−0.11	−0.51 **	0.37 **	−0.19	0.05	0.18	--	
PAS	−0.17	0.12	0.10	−0.20	−0.53 **	0.40 **	−0.17	0.18	0.15	0.47 **	--
T2 CIS	−0.32 *	0.04	0.11	0.12	0.27 *	0.01	0.09	0.10	−0.03	−0.15	0.07

** *p* < 0.01 (significant), * *p* < 0.05 Note. T1-T2 used for all variables unless specified. Neuro-QoL = Quality of Life in Neurological Conditions: Neuro-QoL_social_ = Social Roles and Activities; Neuro-QoL_cog_ = Cognitive Function; Neuro-QoL_well-being_ = Well-Being; Neuro-QoL_fatigue_ = Fatigue; UPDRS = MDS-Unified Parkinson’s Disease Rating Scale; UPDRS-I_non-motor_ = non-motor; UPDRS-II_motor_ = motor; BDI-II = Beck Depression Inventory-II; PAS = Parkinson’s Anxiety Scale; CIS = Coronavirus Impact Scale.

**Table 7 healthcare-13-00070-t007:** Pearson correlation among demographic/clinical characteristics, scores on Neuro-QoL scales, UPDRS scales, BDI, PAS and CIS for women.

	T2 Age	Education	T2 PD Duration	Neuro-QoL_social_	Neuro-QoL_cog_	Neuro-QoL_fatigue_	Neuro-QoL_well-being_	UPDRS_non-motor_	UPDRS_motor_	BDI-II	PAS
T2 Age	--										
Education	0.21	--									
T2 PD Duration	0.10	−0.20	--								
Neuro-QoL_social_	0.09	0.12	−0.14	--							
Neuro-QoL_cog_	−0.01	0.16	−0.19	0.42 **	--						
Neuro-QoL_fatigue_	−0.005	−0.06	0.08	−0.33 **	−0.31 *	--					
Neuro-QoL_well-being_	0.03	−0.04	−0.06	0.38 **	0.17	−0.42 **	--				
UPDRS-I_non-motor_	−0.08	−0.15	0.01	−0.27 *	−0.09	0.20	−0.17	--			
UPDRS-II_motor_	−0.26 *	−0.23	0.24 *	−0.48 **	−0.25 *	0.32 **	−0.42 **	0.31 **	--		
BDI-II	−0.09	−0.22	0.09	−0.23	−0.33 **	0.30 *	−0.46 **	0.16	0.41 **	--	
PAS	−0.23	−0.25 *	−0.09	−0.33 **	−0.29 *	0.25 *	−0.44 **	0.31 **	0.48 **	0.40 **	--
T2 CIS	−0.23	−0.21	0.18	−0.04	−0.15	−0.004	0.20	−0.07	−0.03	−0.07	−0.09

** *p* < 0.01 (significant), * *p* < 0.05 Note. T1-T2 used for all variables unless specified. Neuro-QoL = Quality of Life in Neurological Conditions: Neuro-QoL_social_ = Social Roles and Activities; Neuro-QoL_cog_ = Cognitive Function; Neuro-QoL_well-being_ = Well-Being; Neuro-QoL_fatigue_ = Fatigue; UPDRS = MDS-Unified Parkinson’s Disease Rating Scale; UPDRS-I_non-motor_ = non-motor; UPDRS-II_motor_ = motor; BDI-II = Beck Depression Inventory-II; PAS = Parkinson’s Anxiety Scale; CIS = Coronavirus Impact Scale.

**Table 8 healthcare-13-00070-t008:** Spearman correlation between CIS-items and Neuro-QoL_cog_ and Neuro-QoL_social_ for men.

	Neuro-QoL_social_	Neuro-QoL_cog_	CIS: item1	CIS: item2	CIS: item3	CIS: item4	CIS: item5	CIS: item6	CIS: item7	CIS: item8	CIS: item9
Neuro-QoL_social_	--										
Neuro-QoL_cog_	0.33 *	--									
CIS: item1	0.03	0.06	--								
CIS: item2	−0.09	0.06	0.25	--							
CIS: item3	0.07	−0.01	0.09	0.28 *	--						
CIS: item4	0.05	0.08	0.16	0.18	0.21	--					
CIS: item5	0.27 *	0.30 *	0.16	0.26	0.08	0.31 *	--				
CIS: item6	−0.05	0.01	0.30 *	0.23	0.30 *	0.23	0.08	--			
CIS: item7	0.11	−0.11	0.30 *	0.30 *	0.24	0.02	0.12	0.36 **	--		
CIS: item8	0.18	0.14	0.08	0.03	−0.03	0.20	0.15	0.30 *	0.19	--	
CIS: item9	0.27	0.42 **	0.09	0.18	−0.10	0.09	0.30 *	0.02	0.17	0.34 *	--

** *p* < 0.01 (significant), * *p* < 0.05. CIS = Coronavirus Impact Scale; CIS: item1 = Routines; CIS: item2 = Income/Employment; CIS: item3 = Food Access; CIS: item4 = Medical Healthcare Access; CIS: item5 = Mental Healthcare Access; CIS: item6 = Social Supports; CIS: item7 = Pandemic-Related Stress; CIS: item8 = Family Stress and Discord; CIS: item9 = COVID-19 Diagnosis.

**Table 9 healthcare-13-00070-t009:** Spearman correlation between CIS-items and Neuro-QoL_cog_ and Neuro-QoL_social_ for women.

	Neuro-QoL_social_	Neuro-QoL_cog_	CIS: item1	CIS: item2	CIS: item3	CIS: item4	CIS: item5	CIS: item6	CIS: item7	CIS: item8	CIS: item9
Neuro-QoL_social_	--										
Neuro-QoL_cog_	0.39 **	--									
CIS: item1	0.07	−0.001	--								
CIS: item2	0.11	−0.21	0.06	--							
CIS: item3	0.14	−0.08	0.03	0.18	--						
CIS: item4	0.13	0.04	0.10	0.06	0.26 *	--					
CIS: item5	−0.07	−0.19	0.03	0.19	0.22	0.45 **	--				
CIS: item6	0.04	−0.03	0.17	0.03	0.12	0.17	0.06	--			
CIS: item7	0.06	0.11	0.11	0.03	−0.09	0.16	−0.05	0.23	--		
CIS: item8	−0.05	0.14	0.007	−0.05	0.27 *	0.35 **	0.15	−0.03	0.02	--	
CIS: item9	−0.006	−0.05	0.19	0.02	−0.03	0.07	0.18	0.11	−0.03	0.16	--

** *p* < 0.01 (significant), * *p* < 0.05. CIS = Coronavirus Impact Scale; CIS: item1 = Routines; CIS: item2 = Income/Employment; CIS: item3 = Food Access; CIS: item4 = Medical Healthcare Access; CIS: item5 = Mental Healthcare Access; CIS: item6 = Social Supports; CIS: item7 = Pandemic-Related Stress; CIS: item8 = Family Stress and Discord; CIS: item9 = COVID-19 Diagnosis.

## Data Availability

The original contributions presented in this study are included in the article. Further inquiries can be directed to the corresponding author.

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
