# Peer review of "Changes in Subjective Cognitive and Social Functioning in Parkinson’s Disease from Before to During the COVID-19 Pandemic"

_healthcare, 2025, doi:10.3390/healthcare13010070_

Round 1

Reviewer 1 Report

Comments and Suggestions for Authors

The authors examine changes in quality of life related to cognition and social functioning in people with Parkinson’s disease, over a time span that includes the COVID pandemic (2017/2019 vs 2021). They demonstrate decline in both, which are related to increases in anxiety, fatigue, and motor symptoms. They also report differences between men and women in reported experience with COVID (greater in women), and report a significant correlation in subjective cognition and personal experience with COVID in men.

Overall, the paper is very well-written and addresses important questions related to the indirect effects of the pandemic (lockdowns, societal measures etc.)  on QoL in people with PD. The focus on gender differences and on factors beyond motor symptoms (e.g., cognition, social function, mood) are definite strengths of the paper.  A few elements could be clarified or expanded, as described below.

Specific comments/suggestions:

Introduction: At the end of the section, the authors refer to two studies showing differences between men and women in the impact of COVID, and in how PD affects them (reference 23 and 24).  This should be expanded a bit to provide more information on the nature of these differences.

Statistics/results:

-            While the statistical approach is sound, a few additional details are needed. For instance, parametric statistics are used, but we do not have information on whether assumptions of t-tests and linear regressions are met. This should be reported in the method section. In my experience, clinical scales are rarely normally distributed, and the use of non-parametric or mixed effect models may be better suited in such cases. However, while raw scores at T1 and T2 may not be normally distributed, the change score might.

-            Gender differences:

o   I wonder why gender was not included in the regression models. Given the focus on examining gender differences, if not included in these models, a rationale should be provided.

o   It would be interesting to examine whether the magnitude of the correlations differ between men and women, which cannot be determined by reporting data simply based on whether they pass the statistical threshold. An easy way to do this is to compare correlations using Fisher-z transformations (e.g., https://www.psychometrica.de/correlation.html

Tables:

-            The raw change scores should be reported in Table 1 for ease of comparison with other studies (see below) and to provide the reader with some insight in the degree of change and of variability in this in the sample (i.e., SD).

-            The max score on each scale should be presented in Table 1 to help the reader figure out if scores are at floor or ceiling and whether there is room for these scores to change.

-            Table 5a-5b: CIS items are identified by numbers, but it is unclear what each item measures. The specific correspondence between item and number is also not clearly presented in the methods. This information should be added in both places.

-            It may help to add the subscale number for the UPDRS in the acronym (e.g., UPDRS-I-motor instead of UPDRSmotor) as this may be easily confused with the objective motor exam score on the UPDRS-III subscale.

Discussion:

-            Because disease progression and the pandemic are confounded, it is difficult to determine whether COVID had an additional negative effect on QoL declines. It would be helpful in the discussion to compare the magnitude of the change scores with those documented in other cohorts assessed prior to COVID. For instance, Marras et al (https://doi.org/10.1002/mds.28641) reports changes in the NeuroQol in PD over 3 years, albeit this cohort has a slightly shorted disease duration. Marras has published several papers on QoL in PD, but her work is not included in the manuscript.

-            In the section on cognitive functioning in the discussion, there should be a mention that self-reported depression and subjective cognitive decline often covary across clinical groups including PD. I think following this with the association with objective measures in PD during COVID (ref 38) would make a stronger argument that it is not just a reflection of the negative response-style often observed in individuals with depression

Reviewer 2 Report

Comments and Suggestions for Authors

Overall, I found this manuscript well done, clearly written, and statistically sound, with conclusions that supported the results. One interesting omission was a measure of loneliness. Although the NeuroQoL assesses participation and social connections (objective) based on functional ability, I don't believe (I could be wrong) it effectively evaluates the subjective aspects of loneliness, which literature clearly correlates with poor outcomes. This does not negatively affect my evaluation of the study's value; rather, it is just a consideration for future research. I think a sentence in the limitations about this is warranted.

Line 50, delete "persons with"

Author Response

Overall, I found this manuscript well done, clearly written, and statistically sound, with conclusions that supported the results. One interesting omission was a measure of loneliness. Although the NeuroQoL assesses participation and social connections (objective) based on functional ability, I don't believe (I could be wrong) it effectively evaluates the subjective aspects of loneliness, which literature clearly correlates with poor outcomes. This does not negatively affect my evaluation of the study's value; rather, it is just a consideration for future research. I think a sentence in the limitations about this is warranted.

Response: We have added the requested information (page 13).

Line 50, delete "persons with"

Response: “persons with” has been deleted from Line 50.

Reviewer 3 Report

Comments and Suggestions for Authors

Thank you very much for giving me the opportunity to review this fabulous work. I am adding some comments to facilitate understanding and value of it:

- In summary, it is mentioned that the second assessment is done during the pandemic, in 2021, it would be more appropriate to reflect that it is done after ... . This is repeated in Material and Methods.

- What type of study was done?

- One of the ways of recruiting participants was Clinicaltrials.gov, could this be explained?

- Inclusion and/or exclusion criteria are not included. Parkinson's disease causes cognitive impairment, among other symptoms.

- How was the educational level quantified?

- The Measures section should explain in more detail the variables to be measured and the instruments used for each of them, providing a brief explanation of the relevance and functioning of each one, whether they provide a total score or by dimensions, ....

- Table 1 includes sociodemographic data and data on outcome variables. It should be separated to facilitate understanding

- If the objective of the study is to check the effect of the isolation caused by the pandemic, why does Table 1 show the data of the participants who have not filled out the questionnaires in T2? This information only adds confusion. The data for the participants who have actually been studied should be presented.

- The legend for Table 1 explains the losses. This information should be explained in the results section

- Table 2 should explain in more detail the data it presents.

- Why are the data for the variables at T1 and T2 not shown, and the differences, if any?

- Figure 1 does not appear

- Tables 5 and 6 why do they not present the scores prior to the correlation?

Round 2

Reviewer 3 Report

Comments and Suggestions for Authors

Many thanks for the changes included, which I consider significantly improve the clarity of the work. I would like to make a few minor observations:

Table 1a and 1b are still confusing, showing different groups of subjects that are not clear. The column Only BOSS-PD, what data does it show? What is the difference with All BOSS-PD? It should be explained in the text.

n=214, what does it mean? And why is it presented there?
